# MIMO Radio Frequency Identification: A Brief Survey

**DOI:** 10.3390/s22114115

**Published:** 2022-05-28

**Authors:** Majid Alotaibi, Mohsin Murad, Shakir A. H. Alhuthali, Faisal R. Al-Osaimi, Fahd Aldosari

**Affiliations:** 1Department of Computer Engineering, Umm Al-Qura University, Makkah 21955, Saudi Arabia; mmgethami@uqu.edu.sa (M.A.); frosaimi@uqu.edu.sa (F.R.A.-O.); fmdosari@uqu.edu.sa (F.A.); 2Department of Electrical and Computer Engineering, King Abdul Aziz University, Jeddah 21589, Saudi Arabia; salhuzali0003@stu.kau.edu.sa

**Keywords:** rfid, mimo, survey, multiple antenna, anti-collision, estimation, passive

## Abstract

In this paper, we briefly look at the latest state-ot-the-art in the domain of multi-input multi-output (MIMO) radio frequency identification (RFID) systems while detailing the work done in the domain of anti-collision, range enhancements, bit error rate (BER) improvements and security. Various passive ultra-high frequency (UHF) RFID implementations are considered that employ multiple antennas at the reader and single or multiple antennas at each tag. We look at several recent works those explored MIMO for RFID receivers. When using MIMO at the backscatter channel, significant improvements can be achieved in the BER as well as range extension. With the extra reliability and increased throughput, such systems can be deployed in many important applications like large tag reading scenarios and accurate tracking. Increased throughput is directly dependent on estimation of tag quantity in a bulk reading environment and usually estimators designed for single antenna systems under-perform in such settings causing low signal to noise ratio (SNR) when employed in MIMO systems where tag signal overlapping can happen more often. One of the key challenges is to keep the design of the RFID tag simple, cutting cost and power requirement when employing anti-collision schemes. We provide a brief survey in some of the recent developments related to MIMO RFID systems, the protocols and algorithms used, and improvements achieved.

## 1. Introduction

Radio frequency identification (RFID) allows us to identify objects remotely through the means of wireless communication [1,2]. It has found its application in a variety of domains including but not limited to access control, object identification, transportation, tracking and electronic identification (ID) [3,4,5]. An RFID system is composed of RFID readers and tags along with the RFID software or a middleware application working via application programming interfaces (APIs) [6]. The tag is made up of a small electronic circuit along with an antenna or multiple antennas containing a unique ID. Different types of tags exist including active ones and those that are passive [7]. While an active tag has a power source, usually an internal battery to power the main circuit along with radio frequency (RF) communication, a passive tag relies on power of the incoming signal from the RFID reader’s transmitter and modulates it [8]. As passive RFID tags do not contain an internal power source, they rely solely on the energy of the carrier wave. This causes them to have a short range compared to active tags and finds their applications in short range identification applications. The range can be increased by deploying multiple antennas in an RFID reader but it is limited by the sensitivity of the tag chip [9]. Similarly, tag design can significantly influence the sensitivity and a number of efforts have been undertaken to improve the design [9,10,11,12]. 

Passive tags provide an attractive option for applications where lower costs and a longer shelf life is expected [13]. Various measurements of the channel such as [14,15] proved that the channel for passive RFID systems can be defined as a cascaded one with forward and backscattering components and both channels can be represented via Rayleigh distribution [16]. However, the cascaded channel acts to degrade the transmission since it fades deeper compared to an ordinary Rayleigh channel [17]. Some of the parameters affecting the tag reading ability of the reader are the distance of the tag, its orientation with respect to the receiver, multipath fading, noisy channel, and interference. The interference can be both from multiple tags transmitting signals at the same time or from nearby RFID readers. An RFID system (Figure 1) is usually composed of a receiver with M transmit antennas, N receive antennas, and a tag having L antennas. The core principle behind backscatter communication is such that initially a reader will transmit a continuous wave (CW) to energize the tag or tags. This is followed by the tag sending back the CW signal to the receiver by modifying the impedance loads of its attached antennas according to a particular code prestored in the tag and known as identification data. This technique is referred to as backscatter modulation, whereby the tag transmits secret information to the reader which is modulated over the reflected CW signal. The reader will extract the required information by first estimating and then decoding the signal from the tag after backscatter modulation. The transmission and reception paths, analogous to radar systems, are divided into two categories: bistatic and monostatic [18]. In monostatic systems, a single or multiple antennas first transmit and then listen to the tag response and the channel is half-duplex. On the other hand, in bistatic systems, directional coupling is employed, whereby separate RF channels are used for transmission and reception to enable full-duplex communication. The reader continues to broadcast the CW signal throughout the process to keep the tag energized and at the same time receive the signal echoed by the tag when operating in a full-duplex mode. While such systems are faster, the interference at the receiver side from its own transmit antennas needs to be eliminated. There are various protocols and standards [19] that defines the communication process among the tag and the reader. For HF band, there is ISO 15693 and ISO 18000-3, and ISO 18000-6 and EPCglobal for UHF [20]. Additionally, the UHF frequency range is within 860–915 MHz and it is 13.56 MHz for HF.

Several works have explored multiple-input multiple-output (MIMO) [21] for RFID receivers. When using MIMO at the backscatter channel, significant improvements can be achieved in the bit error rate (BER) as well as extension of reverse-link interrogation range (RIR). With the extra reliability and increased throughput, such systems can be deployed in many important applications like large tag reading scenarios and accurate tracking etc. Increased throughput is directly dependent on estimation of tag quantity in a bulk reading environment and usually estimators designed for single antenna systems underperform in such settings causing average or low signal to noise ratio (SNR) when employed in MIMO systems where tag signal overlapping can happen more often. One of the major areas of recent research has been bulk reading capabilities of the RFID reader systems. To enhance bulk reading capabilities of an RFID system, collision recovery techniques can play an important role as they help in recovering the signal collisions due to interference from neighboring tags who are communicating simultaneously. Most of the recent collision recovery algorithms are based on the assumption that perfect knowledge of the channel state information (CSI) exists but it is difficult to achieve. Block diagram of a typical MIMO RFID system is depicted in Figure 1. The narrowband MIMO channel [22] can be expressed as:(1)$$y\left(t\right)=H^{b}S\left(t\right)H^{f}x\left(t\right)+n\left(t\right),$$
where $H^{b}$ is the channel matrix from the tag to the reader while $H^{f}$ is a channel matrix from reader to the tag, x is the transmitted signal and y is the received signal at the reader. S in Equation (1) represents the backscattering matric and is also known as signaling matrix and n is the noise at receiver. $H^{b}$ and $H^{f}$ are Gaussian distributed and mutually independent. In this work we go through latest state-of-the-art in MIMO RFID systems and briefly look at the tools and techniques used. Some of the major contributions of this work are:A study of various estimation algorithms being employed in case of multi-antenna RFID interrogators.A brief look at various anti-collision algorithms and their viability.A brief look at the recent advancements in the domain of security.

The rest of the paper is organized as follows: Section 2 details various estimation techniques and algorithms used in MIMO RFID systems. Section 3 compares various anti-collision algorithms being used in multi-antenna RFID systems. Section 4 looks at recent advances in the domain of secure MIMO RFID systems. Section 5 presents the discussion and Section 6 concludes the paper.

## 2. Estimation Techniques

Both forward interrogation range (FIR) and RIR are considered when estimating the actual interrogation range of an RFID reader [23]. FIR calculated as the longest distance at which an RFID signal can turn on a tag and is the measure of average power absorbed at its voltage rectifier. RIR, on the other hand, is the longest distance at which the backscattered tag response signal can be accurately detected and is measured using the signal to noise ratio (SNR) that is strong enough for the tag’s data to be read after demodulation. The RIR plays a significant role in evaluating tag reading capabilities of a reader compared to the FIR. The estimation techniques summarized in this work are listed in Table 1. Various estimators employed in the summarized works have been listed along with the frequency range, MIMO configuration and gain. The gain represents performance improvements over a SISO RFID configuration. Kim et al. [24] studies the reverse-link interrogation range of MIMO UHF RFID readers with maximal-ration combining (MRC) and compare it against a SISO reader. The channel is assumed to be arbitrarily correlated to the Nakagami-m distribution. The interrogation range is one of the foremost parameters in evaluating the performance of an RFID reader. Figure 2 details the proposed system architecture of mono-static and bi-static readers. In Figure 2b, it can be observed that from $M_{t}$ transmit antennas, $M_{r}$ receive antennas and a single tag antenna, an $M_{t}\times 1\times M_{r}$ pinhole channel matrix is obtained.

Due to spatial multiplexing, a MIMO based multi receiver antenna-based setup can greatly enhance the BER performance at the receiver [25]. It is derived that using multiple antennas improves the reader SNR and hence improved average RIR. In uncorrelated Rayleigh channels, the gain of a 2×2 MIMO system is 36% higher, while for a 3×3 MIMO system is 60% higher compared to that of a SISO reader system. On the other hand, in correlated environment the gain is almost 26% for 2×2 MIMO and 44% for 3×3 MIMO. The complexity of the system will increase along with hardware and computational costs, but it is worth the improvement.

The indoor channels are usually characterized by a combination of light-of-sight (LOS) component together with dense multipath components (DMC) [26,27]. The authors of [28] propose a UHF MIMO RFID system for high accuracy positioning. A maximum likelihood (ML) direct positioning algorithm for coherent measurements is performed by nearby antennas in a closed setting and non-coherent measurements in long displaced settings. The proposed solution considers the LOS component together with DMCs. The presence of DMCs makes the process of precise positioning more difficult. Exploiting diversity and a higher bandwidth can improve positioning in such channels. An ML positioning technique is used to estimate the position, along with the channel estimates. A dual antenna tag is proposed that supports both 2.45 GHz and UHF bands. The proposed system is able to reduce the position errors below 0.15 m by 80%.

**Table 1 sensors-22-04115-t001:** Comparison of various estimation techniques.

Paper	Algorithm	Frequency Band	Standard	Bandwidth	Tx Power	MIMO	Gain
Kim et al. [24]	Maximal Ratio Combining	917–920.6 MHz	-	600 kHz	30 dBm	3 × 3 2 × 2	36% 60%
Grebien et al. [28]	Maximum Likelihood	865–928 MHz 2.45 GHz	EPCglobal C1 Gen-2	25 MHz	- 8 dBm	4 × 4	80%
Duangsuwan et al. [29]	Minimal Mean Square Error Zero Forcing	2.4–2.5 GHz	EPCglobal Gen-1	-	−20 dBm	2 × 2	20%
Muzamane et al. [30]	Maximum Likelihood	902–928 MHz	Custom MIMO Tag	-	-	1 × 2 × 1	-
Khelladi et al. [31]	Regularized Least Squares	-	EPCglobal C1 Gen-2	-	-	2 × 2 4 × 4	340%
Chen et al. [32]	BABF and Custom Estimator	915 MHz	EPCglobal C1 Gen-2	-	30 dBm	1 × 2, 2 × 1 2 × 3, 3 × 2	90%

While the interest in in the microwave frequencies such as 2.45 GHz is growing rapidly, the authors of [29] make a case for blind signal estimation to detect multiple tags in a MIMO RFID system. The backscattering in RFID interrogation can be modeled as a fading channel, which makes the interrogation process more complex. A perspective of blind channel estimation is presented to the RFID reader receiver for multiuser detection. Multiple antennas have been explored to enhance multiple tag reading capabilities. In the proposed blind estimation-based system, linear equalization techniques have been employed at the receiver including minimal mean square error (MMSE) and zero-forcing (ZF) to estimate the channel state. The channel is assumed to be an indoor one and the proposed blind estimation technique independence component analysis (ICA) is employed for estimating the CIR. The proposed technique outperforms traditional linear estimation schemes when the number of tags is less.

The domain of theoretical modeling of MIMO RFID systems offers a lot of opportunities to study various MIMO techniques in RFID context [33]. Multi-antenna tags enable the possibility of having uncorrelated channel characteristics during the transmission and reception of signals realizing polarization diversity. Muzamance et al. [30] have evaluated the performance of a 1×2×1 MIMO UHF RFID system and propose a new UHF MIMO passive tag prototype and its corresponding reader system. The proposed system consists of a reader with one transmit and receive antenna, whereas the tag has two antennas and is based on a polarization-time coded backscatter diversity scheme. For validation, the experimental BER of the proposed scheme is compared against BER for simulated AWGN, Rayleigh, and Rician models while altering different parameter. For forward and backward channels, a Rician fading model has been used to simulate the chamber room environment. 

One of the key challenges is to keep the design of the RFID tag simple, cutting cost and power requirement when employing anti-collision schemes [34]. Khelladi et al. [31] have proposed a novel channel estimation technique employing concealed pilots for a multi-antenna RFID system. The proposed solution does not use additional data bits and reduces the number of tag interrogation requests and improves channel estimation performance. The technique is based on hidden pilots consisting of a predetermined sequence summed with the data symbols. Since the suggested technique is bandwidth efficient, it enables quick detection of multiple tags. Figure 3 details the proposed architecture with a single transmit and multiple receive antennas. A regularized LS estimation scheme is applied on the preamble part of the tag’s signal that contains the concealed pilots that is same for all tags. The authors also derived solution for the optimization problem and a closed form expression considering the mean square error (MSE) for the given technique. The proposed technique outperforms slotted aloha and post-preamble-based estimation in terms of reading large number of tags. 

In the past decade, various multiaccess algorithms such as slotted ALOHA [35,36] have been proposed to tackle the issue of collision where multiple tags exist in the interrogation range of a reader. To enhance the interrogation throughput, the frame size must be the same length as the number of tags in the interrogation range. However, in a real-life environment, the number of tags in the interrogation range of the reader vary significantly and several works such as [37] have investigated the estimation of tags in the range of an RFID reader. Chen et al. [32] have proposed a multi-antenna RFID system that employs blind adaptive beamforming (BABF) algorithm to improve the interrogation range and the error rate performance in half- and full-duplex modes. Figure 4 [32] depicts the proposed MIMO RFID reader in half duplex and full duplex modes having M antennas. In Figure 4a, both the transmitter and receiver use common antennas leading to full duplex communication, whereas in Figure 4b, separate transmit and receive antennas are used.

In a bistatic reader system, both the forward and reverse links undergo the same channel variations, and the channel coefficients are same, whereas in monostatic systems, they observe independent channel coefficients [38]. The authors have quantified the gains related to reader interrogation range while employing multiple antennas and then used a BABF scheme to improve both the interrogation range and data transmission. The idea is to estimate the antenna weight vectors, by sending a continuous wave to probe the tags, in order to maximize the interrogation range. Simulation results and experimental analysis using Universal Software Radio Peripheral (USRP) device suggests that the proposed scheme complies with the EPCglobal standard and improves the interrogation range as well as the packet error rate of the system.

## 3. Anti-Collision Techniques

An active area of research in MIMO RFID systems is improving the bulk reading capabilities of the reader through improvements in collision recovery [39] from signals of multiple tags. The anti-collision techniques that are reviewed in this work are summarized in Table 2. Various collision recovery algorithms have been used to overcome the collision problem in MIMO RFID systems. It was observed that most of the works used passive tags having EPCglobal Class 1 Gen 2 standard, whereas double Rayleigh fading channel was used in simulations. Salah et al. [40] suggest that considering backscatter link frequency tolerance, which is ignored in traditional collision recovery methods, can be used for enhancing collision recovery. In traditional collision recovery algorithms, backscatter link frequency (BLF) is assumed to be identical for the collided tags. However, since BLF may vary based on manufacturing processes, the tolerance must be considered but it requires perfect channel estimation. Figure 5 details the proposed multi-antenna reader with collision recover. The power spectrum for pilot tones is computed and converted to energy spectrum followed by the proposed metric given in Equation (2). 

By further simulating BLF tolerance artificially, additional improvements can be achieved. The technique is to successfully decode a single tag in a single slot from a group of multiple colliding tags. The power spectrum for each antenna is calculated and then divided by the BLF to be converted into energy spectrum. The peaks can then be used to compute a new metric called Normalized Maximum Energy (NME) for each antenna and given as: (2)$${\mathrm{NME}}_{n_{R}}=\frac{\underset{f}{\max}\left\{E_{s,n_{R}}\;\left(f\right)\right\}}{\sum_{f}\left\{E_{s,n_{R}}\;\left(f\right)\right\}}\;$$
where $E_{s,n_{R}}\;\left(f\right)$ is the energy received on a single antenna $n_{R}$ having a BLF f. The antenna with a higher NME will have a higher SIR and then a single antenna matched filter receiver is utilized for the selected antenna. The proposed technique can improve the bulk reading time by up to 8% over traditional ML receivers, whereas an improvement of 25% can be achieved through the means of artificial rate tolerance simulation. The proposed technique can significantly enhance tag reading in multi-antenna systems without additional complexity overhead.

Collision occurs when the tag signals from multiple tags are received at the same time [47]. In case the number of readers is lower than the number of tags, a multi-label communication model can enhance the interrogation performance through the use of non-sparse schemes [48]. The authors of [41] consider the case of multiple tags, which requires the establishment of a MIMO RFID model. It is assumed that the reader has M antennas while there are N tags that can communicate with the reader. A new method for identifying multiple tags, using blind separation of the source signals based on label collision followed by recovery, is proposed. For bulk reading of tags, an anti-collision algorithm based on Searching-and-Averaging Method in Time Domain (SAMTD) is developed. Furthermore, a two-step extraction method is proposed that considers non-sparse RFID signals. The base vector of the source signal was estimated using the sample of the unit interval. Later, a corresponding signal with smallest interference is extracted in order to avoid a mixed matrix. The throughput analysis suggests that the proposed algorithm outperforms traditional blind source separation (BSS) and ALOHA [49] algorithms. With three antennas, a total of six tags could be perfectly separated. The scheme would work well for scenarios where collision identification is required and where the recovery of the signals is non-fully sparse. 

While existing anti-collision algorithms [50] solely focus on reducing collision probability, they undergo an inherent problem of vast idle slots. Jian et al. [42] propose a collision-tolerant dynamic framed slotted Aloha (CE-DFSA) based technique that is able to identify multiple tags in a slot to decrease the identification time during the identification process. Orthogonal Walsh Sequence (WS) is employed to identify multiple tags that eliminates the need for a spread spectrum. When the RFID reader generates a query, multiple tags would respond having orthogonal WS. Throughput of the system, while considering disparity between the duration of the slots, is not a good estimate of performance evaluation of the anti-collision with regards to the identification time. The authors have thus evaluated their proposed algorithm in terms of time efficiency given as:(3)$$\eta_{\mathrm{time}\_\mathrm{effi}}=\frac{n\;\cdot {\;T}_{\mathrm{ID}}}{I\;\cdot {\;T}_{I}+\;S\;\cdot {\;T}_{S}+\;C\;\cdot {\;T}_{c}}\;$$
where $T_{\mathrm{ID}}$ is the time for transmitting tag ID, S is the successful slots, I represents idle slots, and C is the collision slots whereas $T_{S}$, $T_{I}$, and $T_{C}$ represent the time durations. It is shown via simulations that the proposed technique is able to improve and accelerate the tag identification procedure more efficiently compared to traditional anti-collision algorithms. The proposed scheme is compared against various anti-collision algorithms and the gains have been quantified. 

Multi-antenna RFID schemes are effective in solving the collision problem; however, when employing a quadrature receiver [51], the often-overlooked aspect is that the collided signals can be statistically represented as rectilinear signal with a degree of impropriety. For such improper complex signals, the performance can be improved by processing together the received signal and its complex conjugate using the technique known as widely linear (WL) processing. Deng et al. [43] make a case for a widely linear minimum-mean-square-error (WLMMSE) technique for anti-collision in multi-tag RFID systems. It considers the improper statistics of second order for backscatter tag signals. The proposed technique is effective in separating overlapped tag signals in a single slot and can be extended to multi-antenna RFID systems for throughput enhancements. The simulation and experimental results suggest that the proposed scheme is superior to the traditional signal recovery techniques based on linear models such as linear MMSE (LMMSE).

One of the most prolific standards for RFID systems is the EPCglobal Class-1 Gen-2 and finds its uses in several applications [52]. FM0 encoding scheme used in the EPCglobal Class-1 Gen-2 standard is particularly of interest specially in the domain of logistics. The operating frequency is around 900 MHz, and the range is close to 10 m. Salah et al. [44] derive and evaluate the performance of ML decoding for a UHF RFID system with several receive antennas. The channel is simulated realistically as a doubly selective Rayleigh fading model that makes the decoding process more complex. The performance metrics used are symbol error probability (SEP) and the derived results are compared against experimental simulations. The pairwise error probability (PEP) closes in when the number of antennas increase and gives good results even with two antennas. Perfect channel knowledge was assumed, and it was considered that all tag reply signals undergo block fading.

The throughput of RFID standards such as EPCglobal is restricted by the tag collisions in multi-tag environments [53]. Since the number of tags are unknown at any point in time, they always exceed the frame length that must be equal to the number of tags to be read, hence complicating the performance of the anticollision algorithms. By improving the estimation process of tag quantity, the throughput of an RFID system can be greatly enhanced. Traditional receivers rely on single antenna systems and are usually difficult to be utilized for MIMO RFID systems as the low SNRs due to colliding tag signals degrade their performance significantly. Deng et al. [45] came up with a multi-dimensional tag quantity estimation scheme for MIMO RFID systems that exploits spatial diversity at the multiple antennas of the receiver. The collided tag signals are stored in the form of high-dimensional vectors, thus making the estimation of tag quantity to be modeled as high-dimensional data clusters. It is observed that for a suitable SNR at various backscattering channels, the separation is enhanced by distance increments between clusters enabled by the modeling technique. Therefore, a density-based spatial clustering of applications with noise (DBSCAN) algorithm was integrated into the high-dimensional space to estimate tag quantities. Figure 6 explains the tag recovery algorithm used. The proposed technique outperforms various traditional approaches, and it is proved through simulations and experimentation.

Identification performance in large RFID networks [54] is a key priority and multiple anti-collision techniques have been adopted in the past to enhance this metric. The authors of [46] propose a collision arbitration technique called group-based binary splitting algorithm (GBSA) to tackle the issue of tag collisions through arbitration strategies. The technique is a combination of estimation based on tag cardinality, a smart grouping strategy, and a modified binary splitting algorithm. An ML estimator [55] is employed for computing the cardinality. The tags are categorized into multiple groups according to the cardinality estimation and the optimal grouping design. For the case of multiple tag signals, filling the same time slot, a binary splitting technique is applied to them while the remaining tags are added to a queue and processed later for identification. The system is evaluated using mathematical models followed by simulations where it is observed that the proposed technique outperforms the existing anticollision techniques employed in UHF RFID readers. The use of GBSA technique enhances the system throughput significantly while reducing the complexity of the system.

## 4. Security

RFID is set to become one of the core technologies powering the Internet of Things (IoT) networks [56]. However, with ever growing deployments and the presence of tags everywhere, the privacy concerns are growing. The reader to tag transmission is prone to eavesdropping operations exposing personal information since the backscatter channel is of broadcast nature [57,58]. Several computationally simple cryptography algorithms have been proposed, but they have their limitations. Table 3 lists the few works we summarized associated with security in MIMO RFID systems. In case of a passive tag, eavesdropping attacks are common while active tags are prone to relay attacks. Recently, physical layer security (PLS) [59] techniques have shown themselves as a good alternative to cryptography algorithms as they can ensure proper secrecy and depends upon secrecy capacity. Little work has been done in the domain of backscatter link security employing PLS techniques. Yang et al. [60] investigate the PLS of a MIMO RFID system. A noise injection precoding technique is employed that is ideal for low resourced backscatter channel. Figure 7 details the RFID reader design for EPCglobal C1 Gen-2 tags with the eavesdropper transceiver. A multiantenna tag scenarios is examined with respect to the secrecy rate maximization (SRM) issue [61] by improving both the energy as well as the precoding matrix of the artificial noise at the receiver. The results obtained from the simulations suggest that the proposed technique performs quite well in terms of computational power and enhances secrecy rate of the communication. 

In automotive domain, RFID based keyless entry [62,63] solutions are being adopted at a rapid pace. The passive keyless entry and start (PKES) [62,64] system enables a car to be unlocked after authentication without a physical key. However, this introduces several security challenges and security can be compromised if security of the system is weak making the system vulnerable to attacks. As discussed above, by employing multiple antennas, we can significantly improve the interrogation range of an RFID system, but it will require more sophisticated cryptographic schemes to be developed to enhance the security of the system. Jadoon et al. [65] proposed a sophisticated security technique for PKES systems whereby the reader has multiple antennas to intercept relay attacks. The proposed algorithm is based on HB (Hopper and Blum) [66,67] protocol which is computationally a lightweight algorithm and is modified for use in the proposed MIMO RFID systems. The system employs three antennas at different proximities to deter relay attacks. It is observed through simulation results that the proposed algorithm achieves its intended tasks without compromising the performance of the system while providing protection against relay attacks.

**Table 3 sensors-22-04115-t003:** Comparison of various security schemes for MIMO RFID.

Paper	Algorithm	Layer	Attack	Technique	Rx Antennas
Yang et al. [60]	Projected Gradient	Physical	Passive Eavesdropper	Backscatter	2
Jadoon et al. [65]	Modified HB	Physical	Relay Attack	Challenge Response	3

## 5. Discussion

This paper is an attempt to briefly review the literature related to MIMO RFID systems including academic journals and conference proceedings from the past decade. In doing so, we provide a general outline of the various domains being targeted in the past and present research. With saturation in the traditional RFID systems research and the great potential of multi-antenna systems, more researchers will shift in favor of empirical and theoretical studies in MIMO RFID systems. As explained in the previous sections, the recent MIMO RFID research carried out in the past decade can be divided into three main categories: estimation, anti-collision, and security. The main challenges addressed in the literature were (a) reducing reader to reader interference, (b) bit error rate and capacity improvement through spatial multiplexing, (c) minimizing channel estimation error, (d) collision recovery, and (e) improved security against eavesdropping and relay attacks. While in some papers, authors have developed theoretical frameworks, the work is still in its early phases and further methods and modeling tools need to be designed for evaluating various aspects MIMO RFID systems. It was observed that very few efforts have been undertaken to study the performance improvements of multi-antenna tag designs, which can help improve the sensitivity and range compared to a single antenna tag at the cost of higher cost and power requirements [68]. Similarly, dual-frequency tags (operating both in UHF and GHz band) have only been studied for MIMO RFID systems in [28] and needs to be further evaluated for the enhanced power level allowances in the GHz band. Majority of the articles we reviewed were related to the anti-collision algorithm designs and bulk reading scenarios as this is an area where MIMO RFID systems can improve the gains manyfold. Several algorithms have proposed for improving tag reading times and accuracy. Though many efforts have been carried out to address security issues in RFID systems [69,70,71], further research needs to be conducted in the MIMO context as multiple antenna systems become more prevalent. There are almost no studies on the cost impacts of multi-antenna systems and specific application scenarios where spatial multiplexing can provide higher gains over traditional RFID systems. 

## 6. Conclusions

The state-of-art in MIMO RFID systems was briefly explored and some of the recently proposed algorithms were detailed. We summarized some of the significant improvements achieved through promising new techniques in estimating both the number of tags and fading channel. We have provided a brief survey in some of the recent developments related to MIMO RFID systems, the protocols and algorithms used, and improvements achieved. While doing so, we looked at several recent works that explored MIMO techniques for RFID receivers. Various passive UHF RFID implementations were considered that employed multiple antennas at the reader and single or multiple antennas at each tag. As shown in Table 1, Table 2 and Table 3, a number of recent algorithms have been designed to tackle the issues associated with MIMO RFID systems. One of the key challenges is to keep the design of the RFID tag simple, cutting cost and power requirement when employing anti-collision schemes. The domain of MIMO based communication systems is already quite complex but the computational and power limitations as a unique case of RFID systems makes it more complex. There is a strong need for exploring low cost, low power algorithms for passive UHF RFID systems that are able to enhance various performance metrics without overburdening the system and also stay within the boundaries of the prevalent standards such as EPCglobal Class 1 Gen 2.

## Figures and Tables

**Figure 1 sensors-22-04115-f001:**
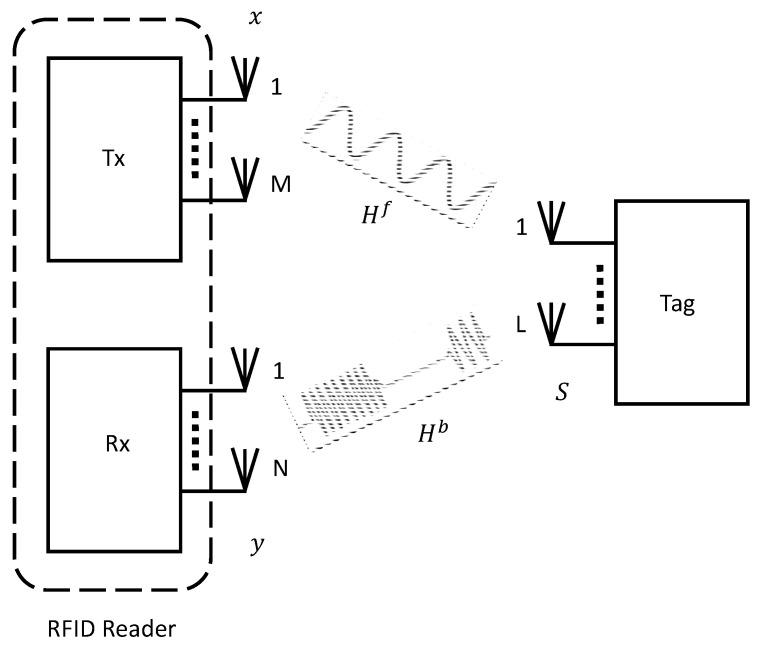
A typical MIMO RFID system.

**Figure 2 sensors-22-04115-f002:**
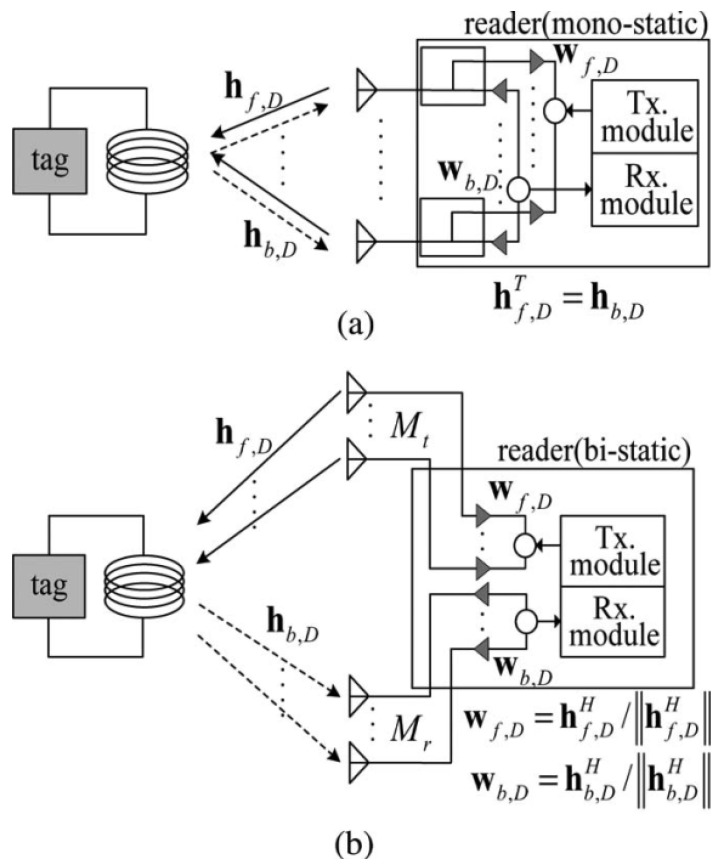
(**a**) Mono-static RFID Reader (**b**) Bi-static RFID Reader. Reproduced from [24] with permission from IEEE.

**Figure 3 sensors-22-04115-f003:**
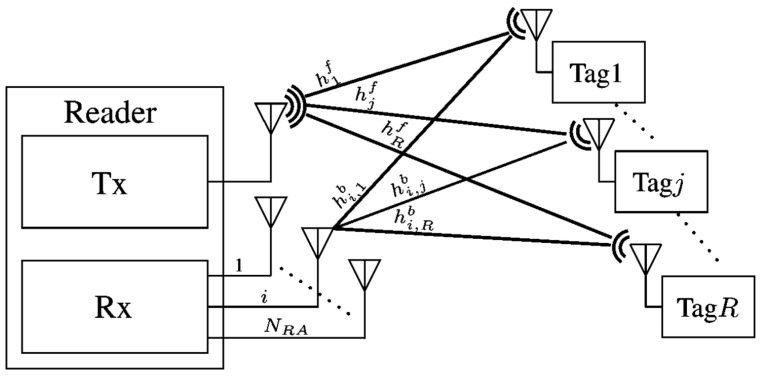
Architecture of the multi receive antenna RFID system. Reproduced from [31] with permission from IEEE.

**Figure 4 sensors-22-04115-f004:**
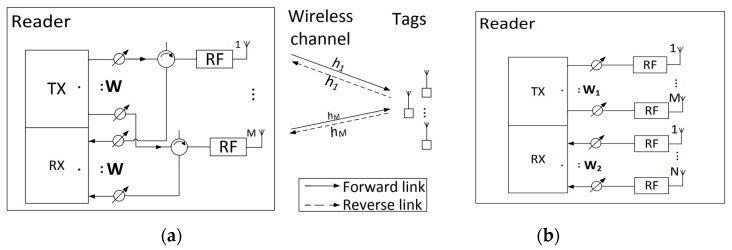
(**a**) Full duplex RFID Reader; (**b**) Half duplex RFID Reader. Reproduced from [32] with permission from IEEE.

**Figure 5 sensors-22-04115-f005:**
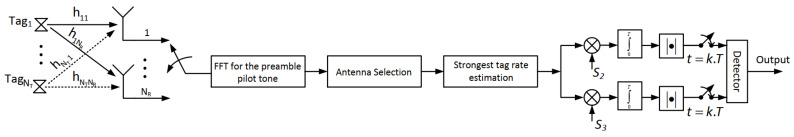
Multi-antenna RFID Reader with collision recovery. Reproduced from [40] with permission from IEEE.

**Figure 6 sensors-22-04115-f006:**
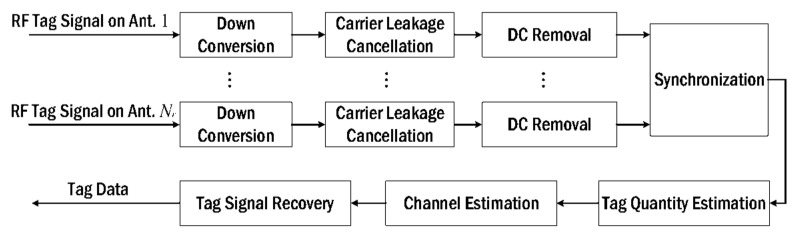
Flowchart of the tag recovery scheme. Reproduced from [45] with permission from IEEE.

**Figure 7 sensors-22-04115-f007:**
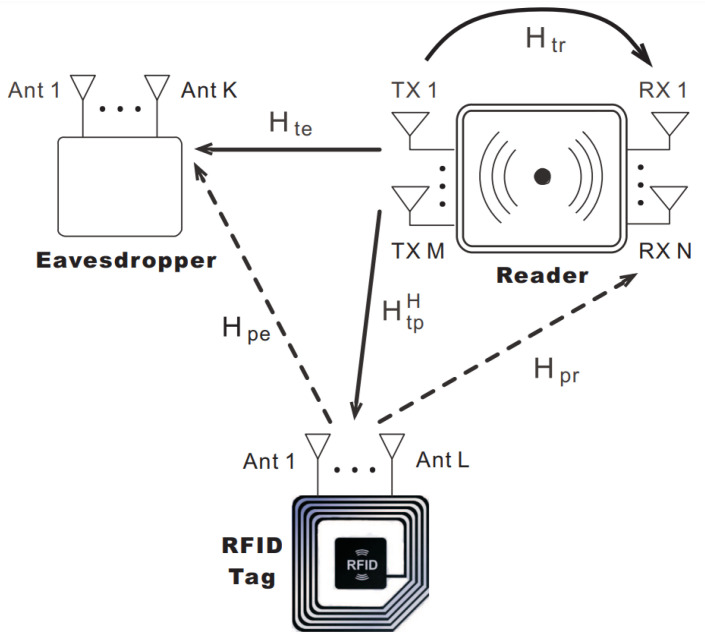
MIMO RFID system with a passive eavesdropper. Reproduced from [60] with permission from IEEE.

**Table 2 sensors-22-04115-t002:** Comparison of various anti-collision techniques.

Paper	Algorithm	Standard	Channel	Rx Antennas
Salah et al. [40]	NME + MF	EPCglobal C1 Gen-2	Double Rayleigh	2, 4
Cheng et al. [41]	Two Step Method	-	Noise	3
Jian et al. [42]	CE-DFSA	EPCglobal C1 Gen-2	Block Fading	-
Deng et al. [43]	WLMMSE	ISO 18000-6C	Quasi-static Rayleigh	4
Salah et al. [44]	ML	EPCglobal C1 Gen-2	Double Rayleigh	2, 4, 8
Deng et al. [45]	DBSCAN	EPCglobal C1 Gen-2	Uncorrelated Rayleigh	4
Su et al. [46]	GBSA	EPCglobal C1 Gen-2	Noise-less	1

## Data Availability

Not applicable.

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
