# Peer review of "MIMO Radio Frequency Identification: A Brief Survey"

_sensors, 2022, doi:10.3390/s22114115_

Round 1

Reviewer 1 Report

The paper summarizes some methods which applied MIMO technique in RFID systems. The performance about anti-collision, range enhancements, bit error rate (BER) improvements and security has been analyzed. Some issues need to be corrected:
1 Some variables are bolded in the expression, but not in the explanation below.
2 More comparison methods need to be added to Table 3.

Author Response

Point 1: Some variables are bolded in the expression, but not in the explanation below.

Response 1: Thank you very much for pointing it out. It was an MS Word error and we have fixed that.

Point 2: More comparison methods need to be added to Table 3.

Response 2: We have updated Table 3.

Reviewer 2 Report

The idea of a brief survey about  "MIMO Radio Frequency Identification" is good. However, this survey was mostly made by using IEEE references. The reviewer strongly suggests that other reference sources must be included like MDPI, ELSEVIER, SPRINGER, RADIO ENGINEERING, ELECTRONICS JOURNAL, HINDAWI, etc. At least, another 20 new references from the afore mentioned must be added. Taking into account that they are based on references of more than 10 years, consider current references.

Describe in the corresponding paragraph what each table interprets, because they do not mention it in the corresponding paragraph. Table 3 is very simple and poor in information, it remains to be completed.

Review in detail the tense of the verbs (past tense/past) participe in the conclusion section.

In lines 189-190, what data are from the comparison?

In line 294, there is one extra brace (parentheses).

You can add about adaptive algorithms (algorithms Q).

Define in each section the types of tags used: EPC Global Class-1 Gen 2 considering which ones are used in each section: EPC Global UHF Class-1 Gen 2 ISO 18000-XX, or all the tags comply with all the characteristics that are presented, if you can add a comparative table where they are considered:

Etimation techniques and algorithms, anticollision and security.

The percentage of narrowband and broadband correlation channels considering bistatic and monostatic, comparative table.

Author Response

Issue 1: The idea of a brief survey about  "MIMO Radio Frequency Identification" is good. However, this survey was mostly made by using IEEE references. The reviewer strongly suggests that other reference sources must be included like MDPI, ELSEVIER, SPRINGER, RADIO ENGINEERING, ELECTRONICS JOURNAL, HINDAWI, etc. At least, another 20 new references from the afore mentioned must be added. Taking into account that they are based on references of more than 10 years, consider current references.

Response 1: Thank you very much for the encouraging remarks. When we started this work, we thoroughly searched the literature for past ten years papers in the context of MIMO RFID systems and most of the literature was from IEEE sources. However, we have now updated the text by citing from other sources as well.

Issue 2: Describe in the corresponding paragraph what each table interprets, because they do not mention it in the corresponding paragraph. Table 3 is very simple and poor in information, it remains to be completed.

Response 2: We have added lines related to each table as suggested. Table 3 has also been modified by adding further information.

Issue 3: Review in detail the tense of the verbs (past tense/past) participe in the conclusion section.

Response 3: We have revised the conclusion section and tried fixing all the grammatic mistakes.

Issue 4: In lines 189-190, what data are from the comparison?

Response 4: The experimental BER curves were compared against theoretical and simulated AWGN, Rician and Rayleigh models for verifications. It has been mentioned in the manuscript as well.

Issue 5: In line 294, there is one extra brace (parentheses).

Response 5: It has been removed.

Issue 6: You can add about adaptive algorithms (algorithms Q).

Response 6: We were unable to find enough works related to Q-protocol in MIMO-RFID context.

Issue 7: Define in each section the types of tags used: EPC Global Class-1 Gen 2 considering which ones are used in each section: EPC Global UHF Class-1 Gen 2 ISO 18000-XX, or all the tags comply with all the characteristics that are presented, if you can add a comparative table where they are considered: Estimation techniques and algorithms, anticollision and security.

Response 7: We have added a tag standard column to Table 1 as well. Previously it was missing. While we could only find a few works related to security, their tag information was provided in their respective paragraphs.

Issue 8: The percentage of narrowband and broadband correlation channels considering bistatic and monostatic, comparative table.

Response 8: We did try to organize a table like that, but not enough data was available in the available papers that we reviewed to make a meaningful table out of it.

Reviewer 3 Report

Check for typos and correctness, e.g.

65: Gen2 is in half duplex

73: range extension only for RIR

77: SNR in Gen2 in average not low since FIR limites coverage

111: eq. (1) no multipath, so limited meaning

130: define RIR

133: "it's" wrong

145: spatial, not special

147/148: RIR range less relevant

Fig 2a. arrows contradictory

Table1: Define "Gain"

182/182: "rapid pace" is wrong

226/227: Monostatic and Bistatic instead of Full and Halfduplex

332: spatial

Author Response

Check for typos and correctness, e.g.

Thank you very much for point these issues out. We have tried to address all the mentioned issues.

Issue 1: 65: Gen2 is in half duplex

Response 1: Here we discussed a generic reader, but we failed to mention both bistatic and monostatic systems. We have updated the text in the revised manuscript.

Issue 2: 73: range extension only for RIR

Response 2: We have updated the text.

Issue 3: 77: SNR in Gen2 in average not low since FIR limites coverage

Response 3: We have updated the text.

Issue 4: 111: eq. (1) no multipath, so limited meaning

Response 4: We have updated the text related to Equation 1.

Issue 5: 130: define RIR

Response 5: We added the definition while addressing Issue 2.

Issue 6: 133: "it's" wrong

Response 6: It has been fixed.

Issue 7: 145: spatial, not special

Response 7: It has been fixed.

Issue 8: 147/148: RIR range less relevant

Response 8: The line has been removed.

Issue 9: Fig 2a. arrows contradictory

Response 9: The figure was adopted from the reference as is.

Issue 10: Table1: Define "Gain"

Response 10: We have defined it in the revised manuscript.

Issue 11: 182/182: "rapid pace" is wrong

Response 11: The sentence was rewritten.

Issue 12: 226/227: Monostatic and Bistatic instead of Full and Halfduplex

Response 12: It has been fixed.

Issue 13: 332: spatial

Response 13: It has been fixed.

Round 2

Reviewer 3 Report

Typos etc are removed but still not a good review of various MIMO RFID approaches. It is a list of papers on MIMO RFID and their individual abstract and conclusions - what is missing is an overview how all these works are interconnected and what are the remaining open questions in this research field.

Author Response

Issue 1: Typos etc are removed but still not a good review of various MIMO RFID approaches. It is a list of papers on MIMO RFID and their individual abstract and conclusions - what is missing is an overview how all these works are interconnected and what are the remaining open questions in this research field.

Response 1: Thank you very much for pointing it out. We have now added a discussion section where we have tried to address the raised concerns including some useful insights on the general direction of the research being carried out and overall limitations.